# Carboxymethylcellulose-Based Hydrogel Obtained from Bacterial Cellulose

**DOI:** 10.3390/molecules28020829

**Published:** 2023-01-13

**Authors:** Sanosh Kunjalukkal Padmanabhan, Leonardo Lamanna, Marco Friuli, Alessandro Sannino, Christian Demitri, Antonio Licciulli

**Affiliations:** Department of Engineering for Innovation, University of Salento, Via Arnesano, 73100 Lecce, Italy

**Keywords:** CMC, bacterial cellulose, hydrogel, rheology, swelling

## Abstract

In the present study, we have produced a sodium carboxymethylcellulose (CMC) hydrogel from a bacterial cellulose etherification reaction with chloroacetic acid in an alkaline medium. Bacterial cellulose (BC) was synthesized via economical and environmentally friendly methods using the *Gluconacetobacter xylinus* bacterium. After purification, freeze-drying, and milling, BC microparticles were dispersed in NaOH solution for different time periods before the etherification reaction. This has allowed the understanding of the alkalinization effect on BC modification. All synthesized CMC were soluble in water, and FTIR and XRD analyses confirmed the etherification reaction. The bath of BC in NaOH solution affects both molecular weight and degree of substitution. SEM analysis revealed the change of BC microstructure from fibrous-like to a smooth, uniform structure. The CMC-0 h allowed the production of crosslinked hydrogel after dehydrothermal treatment. Such hydrogel has been characterized rheologically and has shown a water absorption of 35 times its original weight. The optimization of the CMC produced from BC could pave the way for the production of ultrapure hydrogel to be applied in the healthcare and pharmaceutical industry.

## 1. Introduction

Carboxymethyl cellulose (CMC) is a cellulose derivative with carboxymethyl groups. It is produced by reacting cellulose with monochloroacetic acid (MCA) or its sodium salt in an alkaline medium [1]. Unlike cellulose, CMC is a water-soluble, anionic polysaccharide and is one of the most widely used industrial cellulose ethers [2,3]. CMC is useful in systems where hydrophilic colloids are involved and has many applications in different industrial areas, including cosmetics, pharmaceuticals, food, detergents, textile, etc. [4]. CMC shows the ability to suspend solids in aqueous media, stabilize emulsions, absorb moisture from the atmosphere, and is mainly used as a thickener, suspending aid, binder, gelling agent, stabilizer, and water retention agent, etc. [5,6,7].

Commercial CMC is mostly synthesized from kraft pulp or directly from cotton linters [8]. Over 150 million trees are cut for cellulose production every year, raising issues about the environmental and economical sustainability of such a process [9]. For this reason, several studies have been carried out on utilizing abundant and underutilized cheaper cellulose sources from agricultural waste, such as papaya peel, sugar beet pulp, sago waste, etc., as an alternative to wood for cellulose and CMC production [10,11,12]. Cellulose deriving from plants and waste is a mixture of hemicellulose, lignin, pectin, and other substances and requires a long and not trivial purification process [13,14]. Unrefined commercial-grade CMC is sufficient for the detergents and paper industry, whereas high-purity CMC grades are necessary for the food and pharmaceutical industries [15].

It is well known that bacteria can produce cellulose (BC) which is unique in purity, tensile strength, and chain length. BC, produced by *Gluconacetobacter xylinum,* is an acetic acid bacterium that can ferment carbon sources, converting them to extracellular polysaccharides or cellulose [16]. BC does not require chemical treatments for the removal of hemicellulose and lignin-like impurities and could represent a source for high-quality CMC production. BC consists of a 3D-network structure of nanometer-sized fine fibrils compared to wood-derived cellulose and cotton [17]. The high inherent purity, together with the lower environmental impact on the isolation of bacterial cellulose, encourages its use as an alternative to traditional plant-derived cellulose in CMC production [18]. However, the literature survey showed that there have been only a few reports on the use of BC as a raw material for CMC synthesis [15]. Rachtanapun et al. studied the effects of NaOH concentrations on the degree of substitution (DS) and chemical, physical and thermal properties of CMC and found that NaOH quantity was the major parameter controlling the properties of CMC [4]. Casaburi et al. reported that CMC with DS in the range of 0.60–1.52 (CMC for the food industry) could be successfully synthesized by optimizing the NaOH concentration, molar NaOH/anhydroglucose unit ratio, molar etherifying reagent/anhydroglucose unit ratio, and etherification time [15]. All these works are limited to the characterization of CMC by evaluating DS, structural analysis, viscosity measurements, thermal stability, etc.

In this regard, we tried to explore the possibility of making CMC hydrogel from BC. Together with purity, the profitability of CMC can be raised by modification into a hydrogel. A hydrogel is a hydrophilic polymeric material having a three-dimensional network that can shrink and swell, absorbing a large amount of water. The swelling property of the hydrogel makes it a smart material and is widely used in many applications (e.g., drug release, thickener for food, etc.). The carboxylic group of CMC produces a polyelectrolyte superabsorbent and pH-sensitive hydrogel, and this behavior is described by the Gibbs–Donnan effect. Due to suitable biocompatibility, biodegradability, and lack of taste, pure CMC hydrogels find application in medicine for drug release, patches, medical devices, and also in the food industry [19].

In this work, a systematic study on the alkalization time of BC before adding monochloroacetic acid to synthesize CMC is reported. The obtained CMC samples were characterized for their chemical and physical properties. Moreover, a CMC-based hydrogel has been obtained by exploiting a dehydrothermal crosslinking process (DHT), which preserves the chemical structure of CMC. Finally, water uptake and rheology of such hydrogel have been characterized.

## 2. Result and Discussion

In Figure 1, the XRD pattern of pristine BC and synthesized CMC powders were reported. The crystalline peaks corresponding to cellulose I (2q 14.4 (101), 16.6 (101), and 22.6 (002)) were present in the BC sample [20]. The XRD patterns of CMC samples showed a decrease in crystallinity when the reaction time of BC in NaOH solution increases, caused by the separation of hydrogen bonds in cellulose [21]. The splitting of hydrogen bonds increases the distance between each polymer molecule, which favors the reaction of the monochloroacetic acid molecules into bacterial cellulose. This implies that, before the carboxymethylation reaction, NaOH reaction time affected cellulose structure, decreasing the crystallinity of the CMC.

Figure 2 depicts the FTIR spectra of neat BC and all synthesized CMCs. Typical stretching and vibrational bands of cellulose I appeared for neat BC spectrum at 3405 cm^−1^ for O-H stretching, 2895 cm^−1^ for C-H stretching, 1647 cm^−1^ for H-O-H bending vibration of absorbed water molecules; 1427 cm^−1^ for CH_2_ symmetrical bending, 1164 cm1 for C-O-C bond; 1110 cm^−1^ for C-O stretching and the band at 897 cm 1 for b-linked glucose polymers [22,23]. After the carboxymethylation of BC, the typical absorption bands of carboxymethyl ether groups centered at 1601 cm^−1^ and 1421 cm^−1^ were evidenced, which confirms the formation of CMC [12,24]. The peaks are present in all the CMCs.

The molecular weight and degree of substation of each CMC sample were calculated, and the results are summarized in Table 1. It has been observed that molecular weight decreases with increasing NaOH reaction time with BC, whereas DS does not show any substantial increase with reaction time. The highest molecular weight, 2347 kDa, was obtained for sample CMC-0 h and the lowest (706 kDa) for CMC-12 h. The decrease in molecular weight with increasing reaction time is due to the cleavage of polysaccharides catalyzed by NaOH [25]. There is not much improvement in the etherification of BC with NaOH reaction time observed (Table 1).

The surface morphology of BC and CMC samples was evaluated by scanning electron microscopy and shown in Figure 3a–f. Figure 3a reveals that BC powder is composed of elongated nanofibrous and ribbon-type structures, which is one of the characteristics of bacteria-synthesized cellulose. Figure 3c–f show the morphology of carboxymethylated BC powder at different alkalinization times. It has been observed that in all CMC samples, the structure of starting BC is altered completely after the carboxymethylation reaction, becoming smoother and smoother with increasing time. The complete structural changes found in CMC samples are due to the chemical insertion and attachments of reacting species with the cellulose fiber. The fiber structure of BC has a nanofibrous structure, whereas, after carboxymethylation, the surface structure was lost completely with the loosening of the fibrous structure and conversion into a corroded structure. It can be seen that the corrosion increases with alkalinization reaction time and forms a compact microstructure (Figure 3–f. This structural and morphological change of cellulose after carboxymethylation is considered a result of a reaction with alkali during the modification process [26]. In the case of CMC-12 h, due to excess reaction with alkali, the CMC was obtained in the form of a hard lump instead of powder form and showed a compact microstructure (Figure 3f).

CMC films were subjected to DHT crosslinking in order to obtain crosslinked hydrogel. The only CMC able to crosslink with such treatment was CMC-0h. All other CMC films failed, allowing the production of a viscous gel. This may be due to the decrease in molecular weight with increasing NaOH reaction time (Table 1). Short molecular chains hardly generate a bulk hydrogel but lead to just an increase in viscosity [6]. The swelling capacity of CMC-0 h hydrogels was recorded and reported in Figure 4a. The films swelled 35 times to their original weight in 10 min, and very negligible swelling was evidenced after that. The images of CMC-0h film and corresponding hydrogels are shown in inset Figure 4a. After swelling, the films form a stable and transparent hydrogel, which can hold large amounts of water. Potentially this pure CMC-based hydrogel obtained from BC could find application in healthcare and medical device production (e.g., drug release, cosmetic, etc.), where cellulose purity is important. Figure 4b shows the morphological images of freeze-dried CMC-0h hydrogel. The CMC hydrogel has regular, elongated, and interconnected macropores, having an average size of 600 microns. Pores are indicated by red arrows in Figure 4b. The interconnectivity and pore formation during water absorption leads to the enhanced kinetic and swelling ratio of the hydrogel.

Rheological features of CMC-0h and CMC-1h crosslinked hydrogels are given in Figure 5. The rheological tests highlight the difference between the gels obtained with CMC-0h and CMC-1h. In CMC-0h, the storage moduli (G′) is always higher than the loss moduli (G″), meaning a solid-like behavior [27]. Moreover, the phase angle, which is the time lag between the preset and the resulting sinusoidal oscillation, is inferior to 20°, confirming the generation of a crosslinked hydrogel. On the contrary, the gel obtained with CMC-1h shows G′ and G″ comparable and a phase angle comprised from 40 to 60°. These characteristics describe a viscous fluid state, demonstrating that the CMC was not properly crosslinked. The reduction of the molecular weight together and the increase in carboxyl substitution, driven by the increase in reaction time, prevent the generation of a continuous network for the formation of a strong hydrogel.

## 3. Conclusions

In this study, we successfully synthesized CMC-based hydrogels using bacterial cellulose microparticles. By varying the alkalinization time before the carboxymethylation reaction, we obtained CMC with different molecular weights and degrees of substitution. It has been found that molecular weight drastically reduced when the alkalinization reaction time increased due to the hydrolysis catalyzed by NaOH. The CMC films synthesized without alkalinization before the carboxymethylation reaction allowed the generation of DHT crosslinked hydrogel. The gel exhibited swelling up to 35 times its weight in 10 min, and it was stable for more than 24 h. The rheological characterization of the swelled hydrogel demonstrates the presence of a crosslinking with a phase angle under 20°. The synthesis of CMC-based hydrogel derived from BC could find application in the production of medical devices, where cellulose purity is important; moreover, the utilization of cellulose from a fermentation process could reduce the environmental impact of CMC production.

## 4. Experimental

### 4.1. Materials

Monochloroacetic acid, sodium hydroxide, methanol, and isopropanol (all from Merk) were used as the reagent for the carboxymethylation of BC. For the production of BC, black tea bags (Sir Bolton, Bolzano, Italy), sucrose (commercial product), vinegar (commercial product), and Kombucha fermented tea were purchased from local shops. Sodium hydroxide (NaOH) was purchased from Sigma-Aldrich, Darmstadt, Germany and used without further modification.

### 4.2. Bacterial Cellulose Synthesis

BC pellicles were obtained by the fermentation process of the sweetened black tea with *Gluconacetobacter xilinus* strains, as reported in our previous work [20]. The culture medium was prepared by adding sucrose and tea bags to boiling water (1 L). After removing the tea bags, the pH value of the broth was adjusted to 3 by adding acetic acid. The cellulosic pellicle pieces and liquid broth of the tea fungus (previously activated according to a procedure inspired by the work of Chen and Liu) were added to the cooled tea broth [28]. The fermentation and BC growth were carried at room temperature (28 °C) for 21 days in a static condition. Cellulose hydrogels spontaneously formed on the surface of the broth. These pellicles and the tea fungus were used to inoculate new fermentations. The pellicles were washed with distilled water and soaked in 0.5 M NaOH solution at 80 °C for 120 min under mild shaking conditions to remove any attached cells and impurities. The BC was rinsed in distilled water to remove NaOH, and purified BC pellicles were finally freeze-dried and milled by using a centrifugal mill to obtain the BC nanofiber powder for CMC synthesis.

### 4.3. Carboxymethylation of BC

Carboxymethylation was carried out by a previously reported method with some modifications [18]. For each batch of reaction, 0.5 g of BC powder, 20 mL isopropanol, 1 mL 30% aqueous NaOH, and 0.88 g of chloroacetic acid were used. Firstly, BC powder was suspended in isopropanol in an RB flask and stirred vigorously. Aqueous NaOH was added to the above mixture and stirred for different time intervals (0, 1, 2, 3, 4, and 12 h) at room temperature. After each time interval, chloroacetic acid was added to the mixture, and the RB flask was placed in a water bath at 55 °C for 5 h with stirring. After cooling, the mixture was filtrated, suspended in 30 mL of 80% (*v*/*v*) methanol, and neutralized with acetic acid. The product was washed three times with 30 mL 80% (*v*/*v*) ethanol and subsequently with 300 mL ethanol and dried at 60 °C. The samples were named according to the time in which BC was reacted with NaOH solution; CMC-0 h for 0 min, CMC-1 h for 1 h, CMC-2 h for 2 h, CMC-3 h for 3 h, CMC-4 h for 4 h, and CMC-12 h for 12 h.

### 4.4. Characterization

X-ray diffraction patterns of pristine BC and synthesized CMC samples were recorded using Rigaku Ultima X-ray diffractometer (Rigaku, Tokyo, Japan) using Ni-filtered Cu Kα radiation (λ = 1.5406 Å) at 40 kV. The diffractogram was recorded at 2θ angles between 5° and 45°.

Fourier transform infrared spectra (FTIR) were recorded for dried BC and CMC samples in attenuated total reflectance (ATR) mode with a Perkin Elmer Spectrum One spectrometer (Perkin Elmer, Waltham,) USA at a wavelength range of 400–4000 cm^−1^ with a resolution of 4 cm^−1^. Morphological characterizations of the samples were performed on a Zeiss (Sigma VP, Carl Zeiss, Jena, Germany) field-emission scanning electron microscope (FESEM).

The average molecular weight and polydispersity index of samples of carboxymethylcellulose were determined by gel permeation chromatography (GPC) technique using Agilent GPC/SEC (Agilent, Santa Clara, USA). The CMC solutions for GPC analysis were prepared by dissolving 0.15% *w*/*w* in analytical-grade water. Data were analyzed using an interfaced computer and appropriate data analysis software (Empower3, Waters Corporation, Milan, Italy) to determine Mw and Polydispersity Index.

The degree of substitution (DS) of CMC was determined by a titration method, as described by Bono et al. [29]. A total of 0.5 g CMC was dissolved in 50 mL of demineralized water and 12.5 mL of 0.3 N NaOH solution. The solution was heated to boil for 15 to 20 min. After the products dissolved, the mixture was titrated by 0.3 N HCI. A phenolphthalein indicator was added to observe the color change from Mexican pink (dark pink) to colorless. To calculate the degree of substitution, Equations (1) and (2) were used:(1)$$A=\frac{BC-DE}{F}$$
(2)$$DS=\frac{0.162\times A}{1-\left(0.058\right)\times A)}$$
where:

*DS* = degree of substitution;

*A* = milli-equivalents of consumed acid per gram of specimen;

*B* = volume of NaOH added;

*C* = concentration of NaOH;

*D* = volume of consumed HCl;

*E* = concentration of HCl;

*F* = weight of CMC in grams.

### 4.5. CMC Crosslinking

To obtain a film, 2 wt.% of CMC solutions were prepared in water by stirring at room temperature. The solutions were then cast onto Petri dishes and placed at room temperature for 48 h to obtain dry CMC films. Then, the CMC films were removed from the plates and dehydrothermally crosslinked (0.05 bar at 121 °C for 12 h) in the oven [30].

### 4.6. Swelling Capacity

To study the swelling capacity, the films were immersed in distilled water (at room temperature) for different time intervals, and the degree of swelling was calculated using Equation (3).
(3)$$Q=\frac{wt-wi}{wi}$$
where *wi* is the initial weight of the films in the dry state and *wt* is the weight of the swollen films. The experiment was carried out in triplicate. The synthesis of CMC hydrogel from BC is schematically presented in Figure 6.

### 4.7. Rheological Characterization

The rheological characteristic of both gels was crosslinked and was not assessed by dynamic mechanical test on a parallel plate rheometer (Kinexus Pro+, Malvern, Worcestershire, UK). The tests were performed at 25 °C with the plate surfaces opportunely modified to avoid the hydrogel slipping. A preliminary strain sweep test was performed to define the amplitude region with linear viscoelastic behavior. Frequency sweep measurements were then conducted at constant strain (0.02%) from 0.05 to 10 Hz. The results were recorded in terms of storage (G′) and loss (G″) moduli.

## Figures and Tables

**Figure 1 molecules-28-00829-f001:**
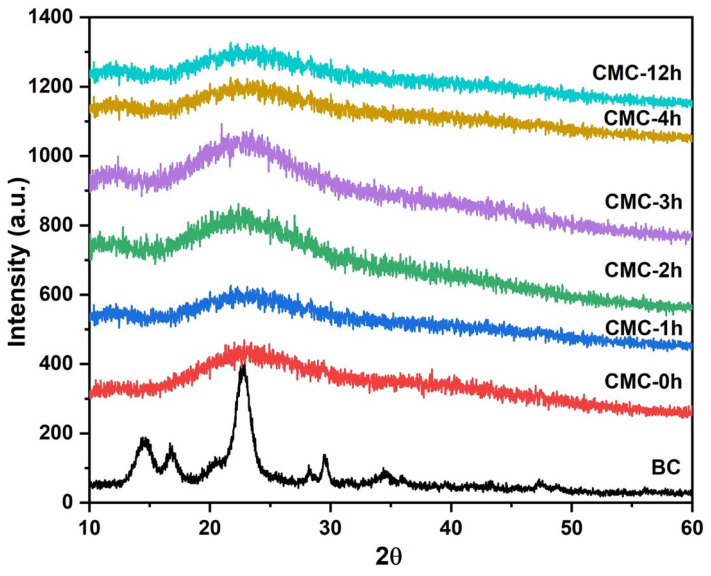
XRD patterns of BC and CMC samples.

**Figure 2 molecules-28-00829-f002:**
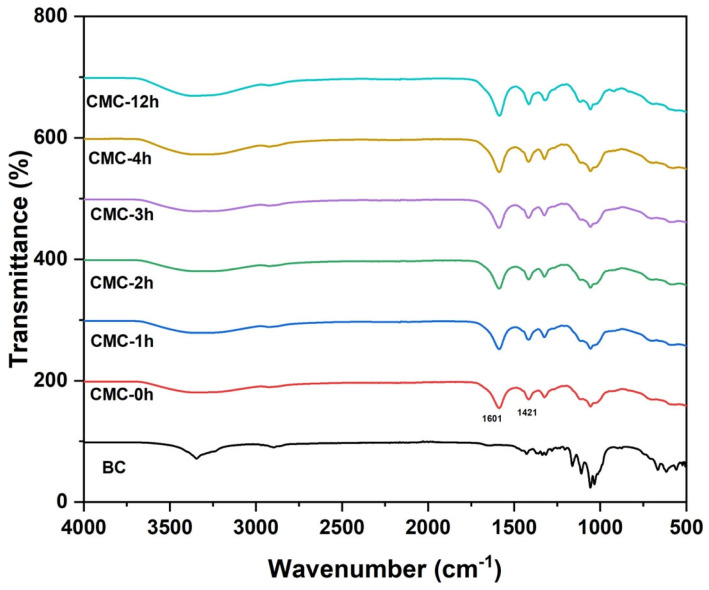
FTIR spectra of BC and CMC samples.

**Figure 3 molecules-28-00829-f003:**
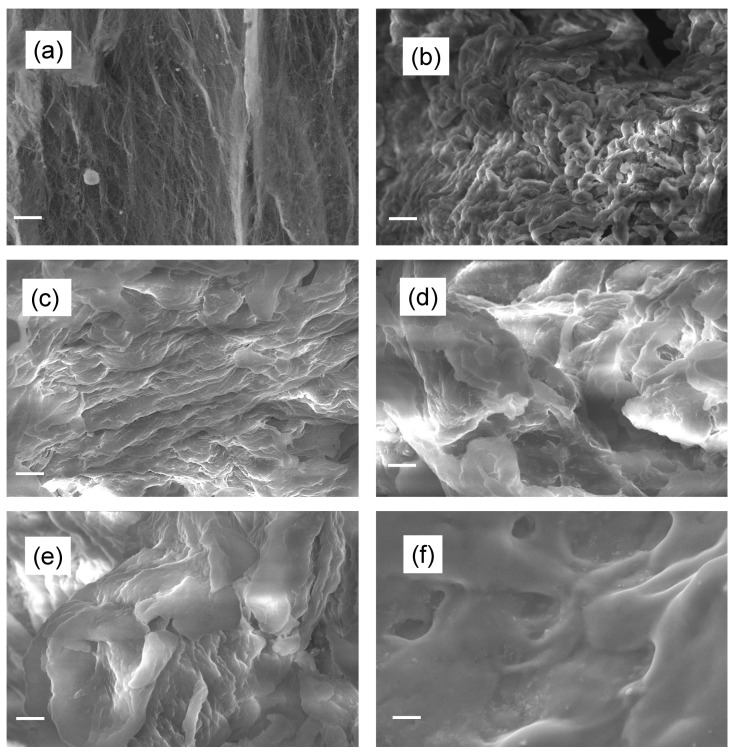
SEM images of (**a**) BC, (**b**) CMC-0 h, (**c**) CMC-1 h, (**d**) CMC-2 h, (**e**) CMC-4 h, and (**f**) CMC-12 h.

**Figure 4 molecules-28-00829-f004:**
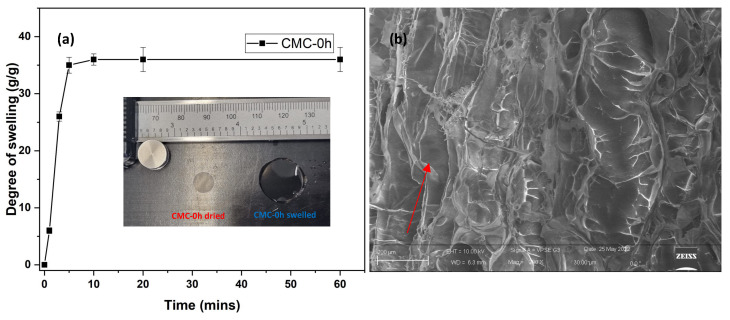
(**a**) Plot representing degree of swelling of CMC-0 h film at different time intervals (inset: image of CMC-0 h film in dry and swelled state) and (**b**) SEM image of freeze-dried CMC-0 h hydrogel (Pores are indicated by red arrows).

**Figure 5 molecules-28-00829-f005:**
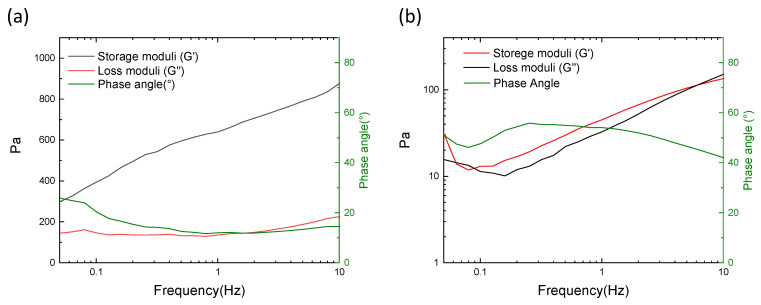
Rheological features of (**a**) CMC-0 h crosslinked hydrogel and (**b**) CMC-1 h hydrogel.

**Figure 6 molecules-28-00829-f006:**
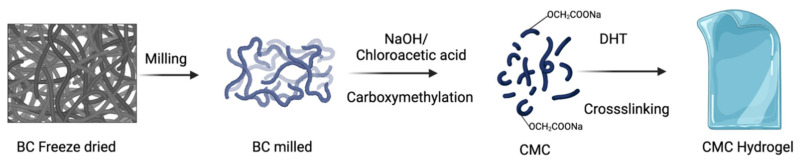
Schematic representation of the synthesis of CMC hydrogel.

**Table 1 molecules-28-00829-t001:** Molecular weight and degree of substitution of CMC samples.

Sample ID	Molecular Weight (kDa)	DS
CMC-0 h	2347 ± 120	0.6 ± 0.02
CMC-1 h	1747 ± 90	0.65 ± 0.02
CMC-2 h	1701 ± 78	0.6 ± 0.05
CMC-3 h	1441 ± 92	0.8 ± 0.1
CMC-4 h	1327 ± 83	0.68 ± 0.08
CMC-1 2	706 ± 40	0.9 ± 2

## Data Availability

Data available on request due to privacy restrictions.

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
