# Peer review of "Carboxymethylcellulose-Based Hydrogel Obtained from Bacterial Cellulose"

_molecules, 2023, doi:10.3390/molecules28020829_

Round 1
Reviewer 1 Report
This work produced CMC from bacterial cellulose via etherification reaction with chloroacetic acid in an alkaline medium and fabricated CMC-based hydrogel according. The work is good-organized. However, there are still several questions needed to be solved before publishment.
1. In Figure 1 and Figure 2, CMC-0h sample which reacted with NaOH solution for 0min showed a dramatic decrease in the cellulose I peak and an increase in the carboxymethyl ether group peak, respectively, demonstrating the successful grafting of the carboxymethyl group. Could you explain why 0min reaction could have that large effect?
2. Please provide detailed information of the DHT method.
3. In Figure 5b, an SEM image of the freeze-dried sample is provided. Why there are no pores formed in the sample?
4. How to conduct the rheological measurement for solid in In Figure 6a?
5. In Page 8, Line 220, you mentioned “The only CMC able to crosslink with such treatment was CMC-0h, all other CMC films failed, allowing the production of a viscous gel.” Why did this phenomenon happen?
6. CMC could be purchased from the vendor in powder form, how to make powder based on your as-prepared sample?
7. What is the difference between plant-cellulose-derived CMC and bacterial-cellulose-derived CMC? The author should provide more data and discussion regarding this issue.
Author Response
Reviewer 1
This work produced CMC from bacterial cellulose via etherification reaction with chloroacetic acid in an alkaline medium and fabricated CMC-based hydrogel according. The work is good-organized. However, there are still several questions needed to be solved before publishment.
We are thankful to you for spending time reviewing our manuscript and for the positive feedback.
- In Figure 1 and Figure 2, CMC-0h sample which reacted with NaOH solution for 0min showed a dramatic decrease in the cellulose I peak and an increase in the carboxymethyl ether group peak, respectively, demonstrating the successful grafting of the carboxymethyl group. Could you explain why 0min reaction could have that large effect?
Thank you for the comments. In the experiment section we have mentioned that after adding NaOH solution and chloroacetic acid, the reaction was carried out for another 5 hr. Our finding was that with an increase in the time of reaction with NaOH solution, the molecular weight of synthesized CMC gradually decreased.
- Please provide detailed information of the DHT method.
Thank you for the comments. the protocol was already written in the experiment section.
“The solutions were then cast onto petri dishes and placed at room temperature for 48 h to obtain dry CMC films. Then, the CMC films were removed from the plates and dehydrothermally cross-linked (0.05 bar at 121°C for 12 hr) in the oven [20].”
- In Figure 5b, an SEM image of the freeze-dried sample is provided. Why there are no pores formed in the sample?
Thank you for the comments. We have pointed out pores in the SEM provided in figure 5b and changed in the text.
“Figure 5b shows the morphological images of freeze-dried CMC-0h hydrogel. The CMC hydrogel has regular, elongated, and interconnected macropores having an average size of 600 microns. Pores are indicated by red arrows in fig 5b.”
- How to conduct the rheological measurement for solid in In Figure 6a?
Thank you for the comments. the protocol was already written in the experiment section.
The rheological characteristic of both gels cross-linked and not were assessed by dynamic mechanical test on a parallel plate rheometer . The tests were performed at 25°C with the plate surfaces opportunely modified to avoid the hydrogel slipping. A preliminary strain sweep test was performed in order to define the amplitude region where with linear viscoelastic behavior. Frequency sweep measurements were then conducted at constant strain (0.02%) from 0.05 to 10 Hz with an applied force of 0.1N. The results were recorded in terms of storage (G’) and loss (G”) moduli.
- In Page 8, Line 220, you mentioned “The only CMC able to crosslink with such treatment was CMC-0h, all other CMC films failed, allowing the production of a viscous gel.” Why did this phenomenon happen?
Thank you for the comments.
We have corrected the text “This may be due to the decrease in molecular weight with increasing NaOH reaction time (table 1). Short molecular chains hardly generate a bulk hydrogel but lead to just an increase in viscosity [6].”
Sited reference https://link.springer.com/article/10.1007/s10570-018-1874-4)
- CMC could be purchased from the vendor in powder form, how to make powder based on your as-prepared sample?
Thank you for the comments. For CMC synthesis, we crushed BC by milling freeze-dried BC. The product obtained after the reaction was in powder form except CMC-12h. CMC-12 was crushed for characterization and other experiments.
Slightly modified the experimental part “The BC was rinsed in distilled water to remove NaOH and purified BC pellicles were finally freeze-dried and milled by using a centrifugal mill to obtain the BC nanofiber powder for CMC synthesis.”
- What is the difference between plant-cellulose-derived CMC and bacterial-cellulose-derived CMC? The author should provide more data and discussion regarding this issue.
Thank you for the comments. It has been well known that, compared to plant-derived cellulose, BC has the advantage of high purity. BC doesn’t have lignin hemicellulose or any other organic impurities found in plant-derived cellulose. This has been clearly mentioned in the introduction part. In this work, our aim was not to compare the CMC from BC and plant-derived cellulose but to study the effect of NaOH activation and synthesis of hydrogel from the obtained CMC.
Reviewer 2 Report
This work reports the preparation of carboxymethylcellulose-based hydrogels from economical and environmentally friendly methods using the Gluconacetobacter xylinus bacterium. The report includes the physicochemical and morphological properties of the hydrogels. In general, the work contains relevant information for the area and is congruent with the “Molecules” journal’s scope. Here are some of the details that need to be addressed before considering this manuscript for publication:
1. Page 1, lines 44-46. “BC, produced by Gluconacetobacter xylinum, is an acetic acid bacterium which can ferment carbon sources, converting them to extracellular polysaccharides or cellulose.” The part marked in bold should be omitted.
2. Figure 3. I suggest leaving only the spectrum for the BC sample and choosing one spectrum for all CMC samples since practically all of them present the same pattern (which can be mentioned in the discussion of the results, and that’s it).
3. Table 1. Please remove the comma in the last row of the second column (706,).
4. Page 6, lines 191-192. The authors mention the following “…whereas DS increases with reaction time.” However, this behavior is not valid for all samples if we look at the data in the Table. Please correct or clarify.
5. Page 8, lines 220-221. The authors mention the following “The only CMC able to crosslink with such treatment was CMC-0h, all other CMC films failed, allowing the production of a viscous gel.”. What causes this behavior? Please explain and support your discussion with some references.
6. There are multiple spelling and grammar mistakes. Please, carefully review the document.
Author Response
Reviewer 2
This work reports the preparation of carboxymethylcellulose-based hydrogels from economical and environmentally friendly methods using the Gluconacetobacter xylinus bacterium. The report includes the physicochemical and morphological properties of the hydrogels. In general, the work contains relevant information for the area and is congruent with the “Molecules” journal’s scope. Here are some of the details that need to be addressed before considering this manuscript for publication:
We are thankful to you for spending time to reviewing our manuscript and for the positive feedback.
- Page 1, lines 44-46. “BC, produced byGluconacetobacter xylinum, is an acetic acid bacterium which can ferment carbon sources, converting them to extracellular polysaccharides or cellulose.” The part marked in bold should be omitted.
Thank you for the comment. It has been corrected.
- Figure 3. I suggest leaving only the spectrum for the BC sample and choosing one spectrum for all CMC samples since practically all of them present the same pattern (which can be mentioned in the discussion of the results, and that’s it).
Thank you for the comment. We want to show by XRD that in all samples the cellulose crystalline peaks are disappeared when forming CMC.
- Table 1. Please remove the comma in the last row of the second column (706,).
Thank you for the comment. Removed comma
- Page 6, lines 191-192. The authors mention the following “…whereas DS increases with reaction time.” However, this behavior is not valid for all samples if we look at the data in the Table. Please correct or clarify.
Thank you for the comments. We have rectified the mistake and modified the text.
“It has been observed that molecular weight decreases with increasing NaOH reaction time with BC, whereas DS doesn’t show any substantial increase with reaction time except for CMC-12h.”
- Page 8, lines 220-221. The authors mention the following “The only CMC able to crosslink with such treatment was CMC-0h, all other CMC films failed, allowing the production of a viscous gel.”. What causes this behavior? Please explain and support your discussion with some references.
Thank you for the comments.
We have corrected the text “This may be due to the decrease in molecular weight with increasing NaOH reaction time (table 1). Short molecular chains hardly generate a bulk hydrogel but lead to just an increase in viscosity [6].”
- There are multiple spelling and grammar mistakes. Please, carefully review the document.
Thank you for the comments. We have rectified spelling and grammatical errors and all changes have been highlighted.
Reviewer 3 Report
The work presented by Sanosh Kunjalukkal Padmanabhan and co-workers is Carboxymethylcellulose -based hydrogel obtained from bacterial cellulose. In general, the authors chose proper research design & methods and appropriated characterization techniques that lead to constructive conclusions. The manuscript is also well prepared from the editorial as well as the scientific point of view. However, I did not see any novelty. The authors should highlight the novelty by comparing it with similar works and telling why this work is important for the reader.
Author Response
The work presented by Sanosh Kunjalukkal Padmanabhan and co-workers is Carboxymethylcellulose -based hydrogel obtained from bacterial cellulose. In general, the authors chose proper research design & methods and appropriated characterization techniques that lead to constructive conclusions. The manuscript is also well prepared from the editorial as well as the scientific point of view. However, I did not see any novelty. The authors should highlight the novelty by comparing it with similar works and telling why this work is important for the reader.
We are thankful to you for spending time to reviewing our manuscript and for the positive feedback.
We have already mentioned in the introduction part about the novelty of our work
“The high inherent purity together with the lower environmental impact on the isolation of bacterial cellulose encourages its use as an alternative to traditional plant-derived cellulose in CMC production[18]. However, literature survey showed that there has been only a few reports on the use of BC as a raw material for CMC synthesis[15]. Rachtanapun et. al. studied the effects of NaOH concentrations on the degree of substitution (DS), and chemical, physical and thermal properties of CMC and found that NaOH quantity was the major parameter controlling the properties of CMC [4]. Casaburi et.al reported, CMC with DS in the range of 0.60-1.52 (CMC for the food industry) could be successfully synthesized by optimizing the NaOH concentration,molar NaOH/anhydroglucose unit ratio, molar etherifying reagent/anhydroglucose unit ratio, and etherification time[15]. All these works are limited to the characterization of CMC by evaluating DS, structural analysis, viscosity measurements, thermal stability, etc. In this regard, we tried to explore the possibility of making CMC hydrogel from BC. Together with purity, the profitability of CMC can be raised by modification into a hydrogel.”
Round 2
Reviewer 1 Report
The authors have already revised the manuscript according to comments. I consider this manuscript as worthy for publication in Molecules.